# The ubiquitin ligase PHR promotes directional regrowth of spinal zebrafish axons

Juliane Bremer [1], Kurt C. Marsden [1,3], Adam Miller [2] & Michael Granato[1]

To reconnect with their synaptic targets, severed axons need to regrow robustly and directionally along the pre-lesional trajectory. While mechanisms directing axonal regrowth are poorly understood, several proteins direct developmental axon outgrowth, including the ubiquitin ligase PHR (Mycbp2). Invertebrate PHR also limits regrowth of injured axons, whereas its role in vertebrate axonal regrowth remains elusive. Here we took advantage of the high regrowth capacity of spinal zebrafish axons and observed robust and directional regrowth following laser transection of spinal Mauthner axons. We found that PHR directs regrowing axons along the pre-lesional trajectory and across the transection site. At the transection site, initial regrowth of wild-type axons was multidirectional. Over time, misdirected sprouts were corrected in a PHR-dependent manner. Ablation of *cyfip2*, known to promote F-actin-polymerization and pharmacological inhibition of JNK reduced misdirected regrowth of PHR-deficient axons, suggesting that PHR controls directional Mauthner axonal regrowth through *cyfip2*- and JNK-dependent pathways.

[1] Department of Cell and Developmental Biology, Perelman School of Medicine, University of Pennsylvania, Philadelphia 19104 PA, USA. [2] Institute of Neuroscience, University of Oregon, Eugene 97405 OR, USA. [3] Present address: Department of Biological Sciences, North Carolina State University, Raleigh 27607 NC, USA. Correspondence and requests for materials should be addressed to J.B. (email: juliane@bremer.ch)

After injury, severed central nervous system (CNS) axons need to regrow robustly and directionally along the pre-lesional trajectory to reconnect with their targets. In the adult mammalian vertebrate CNS, nonpermissive neuron-intrinsic and extracellular milieus prevent axons from regrowing beyond the lesion site[1]. Recently, mammalian CNS axonal regrowth beyond the lesion site has been achieved by manipulating neuron-intrinsic factors, including developmentally active transcription factors[2,3], cAMP[4], and PTEN/mTOR[5–7]. Combined activation of mTOR and JAK/ STAT3 pathway[8] or cAMP and Neurotrophin-3[9] further enhanced axonal regrowth. Strikingly, a conditioning lesion combined with spatially controlled neurotrophin-3 expression was able to direct regrowing CNS sensory ascending axons to their appropriate brain stem target in mouse[10]. Nevertheless, to date there is only limited insight into the mechanisms controlling extent and direction of regrowing CNS axons after injury.

While the mechanisms controlling CNS axonal regrowth after injury are only partially understood, several genes are known to control and direct growing axons during development. One of these genes is the ubiquitin ligase Pam/Highwire/Rpm-1 (PHR) protein. The PHR protein family is composed of highly conserved multidomain proteins. Except for the human orthologue Pam (Protein associated with Myc) that binds the transcription factor Myc[11], PHR protein family members have been identified in independent forward genetic screens for genes controlling developmental synapse formation, axon outgrowth and guidance. These include *highwire (hiw)* in drosophila[12], *rpm-1* in *Caenorhabditis elegans*[13,14], *esrom*, also designated *MYC binding protein 2 (mycbp2)* in zebrafish[15,16], and *Phr1* in mouse[17,18]. Invertebrate PHR controls synaptic and axonal development by downregulating the growth-promoting MAP kinase kinase kinase (MAP3K), known as dual leucine zipper-bearing kinase (Dlk) through ubiquitinating Dlk and targeting it for degradation by the proteasome[19–21]. In addition to its developmental role, invertebrate PHR also limits axonal regrowth after injury[22–24] by negatively regulating Dlk[19,20].

Despite the highly conserved role in synaptic and axonal development, the functions of vertebrate and invertebrate PHR differ substantially. In contrast to invertebrate PHR[12–14], vertebrate PHR is required for the formation of major axon tracts in the CNS and for animal survival[17,25]. Furthermore, while PHR-deficient invertebrate axons showed aberrant branching and overgrowth beyond the target[12,13,26], PHR-deficient vertebrate axons were not only misguided but frequently stalled in mice[18] and zebrafish[27]. In addition, Dlk is not the only PHR effector in vertebrates[25,28,29]. Hence, the PHR protein family has highly conserved functions in neuronal development, which partially differ in downstream targets and functional consequences. While the growth-promoting effect of Dlk on axonal regrowth is evolutionarily conserved[22–24,30–32], the role of PHR in vertebrate axon regeneration remains elusive[33].

Fish have proven to be a valuable model system to study vertebrate axonal regrowth. In contrast to mammals, most fish CNS axons can regrow after injury[34]. Although selected fish CNS axons, including the Mauthner axon, show defective regeneration after spinal cord transection[34–36], increasing intraneuronal cAMP levels[35] or spatially more precise 2-photon-laser-mediate transection enabled spinal Mauthner axons to regrow after injury[37]. Here, we took advantage of the high regrowth capacity of larval zebrafish Mauthner axons after laser-mediated axonal transection to identify vertebrate genes controlling extent and direction of CNS axonal regrowth. Using a candidate gene approach, we identified four genes that were required to promote CNS axonal regrowth along the pre-lesional trajectory. All four genes controlled the extent of caudally directed CNS axonal regrowth, including (1) the noncanonical wnt-signaling component

cadherin EGF LAG Seven-Pass G-type receptor 3 (*celsr3*), (2) *dynein cytoplasmic 1 heavy chain 1* (*dync1h1*), an essential component of dynein which mediates retrograde transport, and (3) *cytoplasmic FMR1 Interacting Protein 2* (*cyfip2*), a component of the WAVE complex that promotes F-actin polymerization. The fourth gene, *mycbp2* that encodes the ubiquitin ligase PHR promoted directional regrowth of spinal Mauthner axons. We found that regrowing wild-type axons crossed the transection site by first initiating multidirectional sprouting and then correcting misdirected sprouts in a PHR-dependent manner. PHR was required to guide regrowing axons, not only across the transection site, but also along the pre-lesional trajectory. Furthermore, we found that PHR controlled F-actin dependent features of growth cones of regrowing axons. Ablation of *cyfip2*, known to promote F-actin-polymerization as well as pharmacological inhibition of JNK reduced misdirected regrowth of PHR-deficient axons, suggesting that PHR controls directional Mauthner axonal regrowth through *cyfip2*- and JNK-dependent pathways.

## Results

**Severed Mauthner axons regrow robustly and directionally**. To identify genetic regulators of vertebrate CNS axonal regrowth, we developed an assay to laser transect the Mauthner axon and visualize axonal regrowth in live larval zebrafish. In larval zebrafish, Mauthner neurons have large-caliber axons that extend along the ventral spinal cord and can be labeled transgenically and easily visualized (Fig. 1a, b)[38]. We chose to study regrowth of the larval Mauthner axon, because it shares crucial properties with the mature mammalian CNS. In the mature mammalian CNS, axons are myelinated by myelin basic protein (MBP)-positive oligodendrocytes and have synaptic connections with their target cells. Both myelination and synaptogenesis correlate with the developmental decline in CNS axonal regrowth capacity[1,39]. In addition, mammalian CNS axons are embedded in a glial matrix composed of glial fibrillary acid protein (GFAP)-positive astroglial cells, which affect CNS axonal regrowth[1]. Similar to mammals, Mauthner axons in 5-day-old larval zebrafish Mauthner are surrounded by myelinating MBP-positive oligodendrocytes and GFAP-positive glial cells (Fig. 1c, d)[40,41] and have functional synaptic connections with downstream neurons that mediate short-latency escape behavior[42]. The mammalian-like cellular composition of larval zebrafish combined with the mature features of the Mauthner neuron and the ease of visualizing the large-caliber axon establish the Mauthner axon as an ideal model for studying CNS axonal regrowth.

We used a laser on a spinning disc confocal microscope to transect the Mauthner axon in live larval zebrafish (Fig. 1e–k). Time-lapse imaging showed that within 1 h post transection (hpt), the proximal and distal axonal stumps retracted slightly and resealed, leaving a gap of $40.1 \pm 13.9 \, \mu m$ (Fig. 1f, Supplementary Movie 1). The distal axonal stump remained intact for several hours and then underwent stereotyped fragmentation known as Wallerian degeneration[43,44], at $28.1 \pm 7.6$ hpt (Fig. 1g, h, Supplementary Movie 2), similar to previous reports[37,45]. By 15 hpt, the proximal axon stump sprouted in rostral, caudal, ventral, and dorsal directions at the transection site in seven of eight axons (Fig. 1i). Over time, misdirected trajectories were corrected in all seven axons and the regrowing proximal axon crossed the transection site to regrow caudally. By 48 hpt, all eight axons (one axon per larva) regrew between 532 and 1063 $\mu m$ caudally to a mean length of $789 \pm 194 \, \mu m$ beyond the transection site (Fig. 1j, k). Thus, laser-mediated transection of the Mauthner axon combined with confocal imaging enabled us to observe the cellular processes during CNS axonal degeneration and directional regrowth in real time.

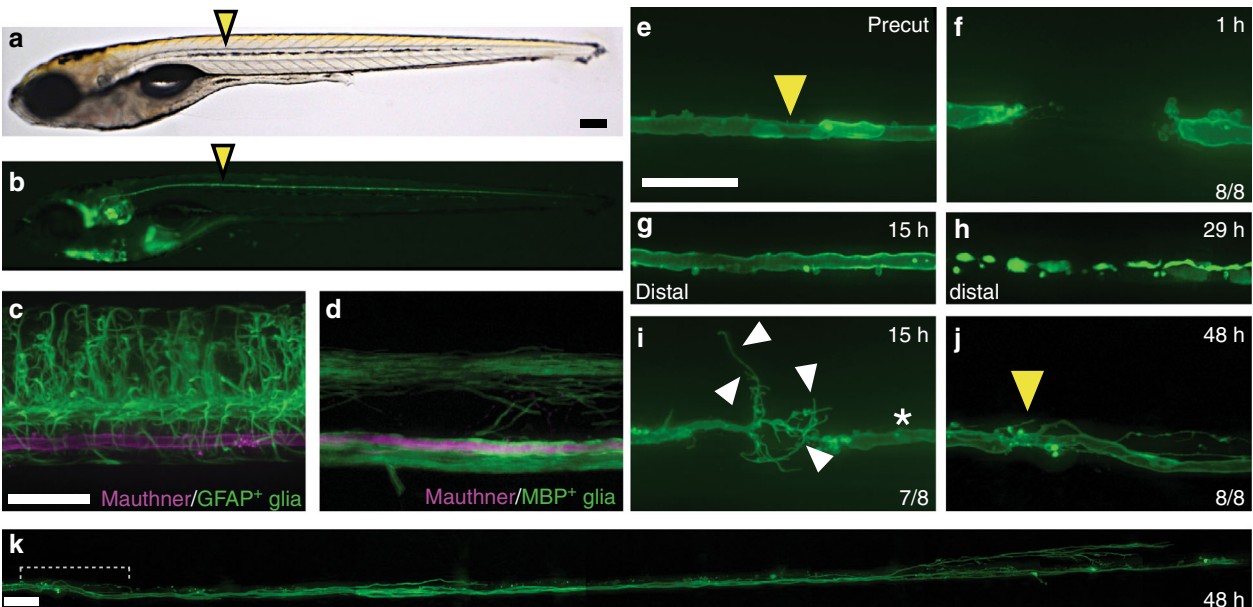

**Fig. 1** Robust and directional regrowth of Mauthner axons following laser-mediated transection. **a, b** 6-day-old double transgenic zebrafish larvae with labeling of the Mauthner neuron: *Tg(hspGFF62a)* and *Tg(UAS:gap43_{1-20}-citrine)*, driving expression of membrane-bound citrine in the Mauthner using the Gal4-UAS system; bright field in (**a**), citrine labeled Mauthner axonal membrane along the ventral spinal cord in (**b**). Yellow arrowheads mark the future axon transection site at the ninth body segment. **c, d** In 5-day-old larval zebrafish, the Mauthner axon, labeled by *Tg(hspGFF62a)* and *Tg(UAS:gap43_{1-20}-RFP)* is surrounded by GFAP-positive glial cells in *TgBAC(GFAP:GFAP-GFP)* (**c**) and myelinating oligodendrocytes in *Tg(MBP:EGFP-CAAX)* (**d**). **e–k** Laser-mediated transection of the Mauthner axon, intact axon before transection with the future transection site marked by a yellow arrowhead (**e**); retracted and sealed proximal and distal axon stump at 1 hpt (**f**), both at the ninth body segment. The axon stump distal to the transection site (here at the level of the anus) remains intact for several hours after transection (**g**) and then undergoes Wallerian degeneration, leaving behind the fragmented axon (**h**). Of eight axons, one fragmented earlier than 13.5 hpt and another later than 36 hpt. The mean time of fragmentation at the level of the anus for the remaining six axons was 28.1 ± 7.6 hpt. At 15 hpt, the distal stump was still intact in this example (white asterisk) and the proximal stump had started to regrow (**i**). Initially, the Mauthner axon regrew multidirectionally in seven of eight larvae, with some sprouts pointing caudally across the transection site; others pointed rostrally toward the cell body in the brain stem (white arrowheads; **i**). **j, k** At 48 hpt, the regrowing proximal axon had successfully crossed the transection site (yellow arrowhead). Misdirected axonal projections at the transection site were corrected (**j**) and the axon had regrown caudally. The entire length of the regrown axon is shown in (**k**). Scale bars: 200 µm in (**a**); 30 µm in (**c**), 30 µm in (**e**), and 30 µm in (**k**)

**Celsr3, Dync1h1, Cyfip2, PHR control extent of axon regrowth.**
After establishing an assay to visualize directionally regrowing CNS axons in live vertebrates, we sought to identify genes controlling the extent and direction of Mauthner axonal regrowth. We selected 15 genes with known function and/or expression in the nervous system and tested whether these genes control the extent and/or direction of Mauthner axonal regrowth after transection. At 5 days post fertilization (dpf), we transected the Mauthner axon in genetic mutants of those genes and their nonmutant siblings to determine the extent of regrowth at 48 hpt. Eleven genes did not obviously affect a Mauthner axon regeneration (Table 1).

However, we identified four genes that were required for the extent of caudally directed Mauthner axonal regrowth (Fig. 2 and Table 1): *cadherin EGF LAG seven-pass G-type receptor 3* (*celsr3*; Fig. 2c, h), *dynein cytoplasmic 1 heavy chain 1* (*dync1h1*; Fig. 2d, h), *cytoplasmic FMR1 interacting protein 2* (*cyfip2*; Fig. 2e, h), and the ubiquitin ligase *mycbp2* (hereafter referred to as *phr*; Fig. 2f–h). While *celsr3*-deficient axons regrew significantly less than in nonmutant siblings, all *dync1h1*-deficient (5/5) and most *cyfip2*-deficient axons (6/8) completely failed to form a growth cone and most *cyfip2*-deficient axons (7/8) failed to regrow beyond the transection site. Interestingly, in all examined *dync1h1*-deficient axons, the proximal axon stump at the transection site became enlarged, in line with accumulating anterogradely transported material and defective retrograde transport[46], while failing to form a growth cone (Fig. 2d). In contrast, transected *cyfip2*-deficient axons without a growth cone formation displayed a pointed proximal stump (Fig. 2e).

Compared to axons in nonmutant siblings, PHR-deficient axons in *mycbp2^{tn207b/tn207b}* (hereafter referred to as *phr* mutants or *phr^{-/-}*) showed slightly less caudal regrowth, demonstrating that PHR promotes caudally directed Mauthner axonal regrowth (Fig. 2f–h). The role of PHR in controlling axonal regrowth was not restricted to spinal axons, since we also observed significantly reduced axonal regrowth in peripheral motor nerves of *phr* mutants (Supplementary Fig. 1). In summary, using a candidate gene approach, we identified four genes that control the extent of caudally directed Mauthner axonal regrowth: *celsr3, dync1h1, cyfip2,* and *mycbp2 (phr)*.

**PHR corrects misdirected sprouts to direct axonal regrowth.**
Careful assessment of the morphology of regrown Mauthner axons revealed rostrally misdirected axonal trajectories in 63% of all regrown PHR-deficient axons (*n* = 37) compared to only 8% of nonmutant sibling axons (*n* = 27), suggesting that PHR was required to direct axonal regrowth across the transection site (Figs. 2g and 3a–f, j). In addition, compared to nonmutant siblings, PHR-deficient axons showed significantly more ventrally misdirected trajectories emerging from the spinal cord (Figs. 2f and 3g, k). Furthermore, significantly more PHR-deficient axons regrew caudally, but followed a path along the dorsal spinal cord instead of the pre-lesional trajectory along the ventral spinal cord (Fig. 3a–e, h–i, l). Together these findings demonstrate that PHR is required for directional regrowth across the transection site and along the pre-lesional trajectory.

**Table 1 Genes/alleles tested and effects on axon regeneration observed in this study**

| Gene | Allele | Mutation | Axon regeneration |
|---|---|---|---|
| *cfl1l* | sa5863[80] | Essential splice site mutation, affecting AA 1 of 163 | Normal Mauthner axonal regrowth |
| *dicer1* | hu896[81] | Premature stop codon at AA 172 of 1865 | Normal Mauthner axonal regrowth |
| *dgcr8* | fh344[78] | Premature stop codon at AA 139 of 782 | Normal Mauthner axonal regrowth |
| *slow learner* | p174[82] | n.d. | Normal Mauthner axonal regrowth |
| *lrp4* | p184[83] | Frame shift mutation at AA 319, premature stop codon at AA 331 of 1899 | Normal Mauthner axonal regrowth |
| *nf1a* and *nf1b* | *nf1a*$^{\Delta5}$ and *nf1b* +1032 | $\Delta5$ bp indel causing a premature stop codon at AA 1075 of 2805 (*nf1a*) or a 10 bp insertion, causing a premature stop codon at AA 665 of 2729 (*nf1b*) | Normal Mauthner axonal regrowth |
| *pappaa* | p170[82] | Premature stop codon at AA 322 of 1591 | Normal Mauthner axonal regrowth |
| *rb1* | te226a[84] | Splice site mutation, premature stop codon at AA 642 of 903[94] | Normal Mauthner axonal regrowth |
| *robo2* | ti272z[85] | Premature stop at AA 635 of 1513[95] | Normal Mauthner axonal regrowth |
| *sox10* | m241[86] | n.d. | Normal Mauthner axonal regrowth |
| *tnc* | sa1576[80] | Premature stop codon at AA 990 of 1811 (does not affect all transcripts) | Normal Mauthner axonal regrowth |
| *celsr3* | fh339[78] | Premature stop codon at AA 215 of 2065 | Significantly reduced Mauthner axonal regrowth |
| *dync1h1* | hi3684Tg[79] | Transgenic insertion into the 52nd intron, affecting AA 3397 of 4548 | No Mauthner axonal regrowth |
| *cyfip2* | p400[77] | Premature stop codon at AA 342 of 1253 | Strongly and significantly reduced Mauthner axonal regrowth |
| *mycbp2* (*phr*) | tn207b and tp203[15] | tn207b: premature stop codon at AA 3413 of 4574 tp203: splice donor site mutation, causing premature stop at AA 3372 of 4574[27] | tn207b/tn207b and tn207b/tp203: reduced extent and abnormal directionality of Mauthner axonal regrowth; tn207b/tn207b: reduced extent of peripheral nerve regrowth |

*AA* amino acid, *n.d.* not determined

In contrast to *phr* mutants, heterozygous *phr*^tn207b/+^ larvae did not display reduced extent of caudally directed axonal regrowth or an increased percentage of rostrally misdirected trajectories compared to wild-type larvae (Fig. 4). This argues against haploinsufficiency of *phr* or a dominant negative effect of the *phr*^tn207b^ allele. To confirm that the axonal regrowth deficits in *phr* mutants are caused by a disruption of the *phr* gene, we tested whether a second *phr* allele called tp203 would complement the tn207b allele. In contrast to Mauthner axons in heterozygous *phr*^tn207b/+^ larvae, axons in compound heterozygous *phr*^tn207b/tp203^ larvae displayed a reduced extent of caudally directed axonal regrowth and significantly more rostrally misdirected trajectories, similar to our previous observations in *phr*^tn207b/tn207b^ (Fig. 4). This demonstrates that it is indeed *phr*, which promotes extent and directionality of Mauthner axonal regrowth.

Since intact distal axon stumps can be inhibitory to axonal regrowth[47] and a delay in Wallerian degeneration has been reported in PHR-deficient mice and drosophila[24,48], we determined whether a delay in Wallerian degeneration could impair axon regrowth in *phr* mutant zebrafish larvae. In peripheral motor nerves we found that *phr* mutant axons fragmented 37% later than nonmutant sibling axons (full fragmentation after transection in nonmutant siblings at $244 \pm 35$ min; in *phr* mutants at $336 \pm 35$ min; Supplementary Fig. 1). However, since the first regrowing motor axon sprouts emerge from the proximal stump several hours later (around 9 h or 540 min after transection)[49], reduced regrowth of *phr* mutant peripheral motor nerves cannot be caused by the intact distal stump. In *phr* mutant CNS Mauthner axons however, we did not observe a delay in Wallerian degeneration. At 17 hpt 4/7 *phr* mutant axons and 4/8 axons in nonmutant siblings showed at least partial fragmentation and at 48 hpt 29/29 *phr* mutant axons and 39/39 axons in nonmutant siblings showed complete fragmentation (Supplementary Fig. 2). Therefore, PHR does not control extent or directionality of Mauthner axonal regrowth by promoting axon fragmentation in larval zebrafish.

The aberrant multidirectional trajectories of PHR-deficient axons could be explained by a role for PHR in limiting general axonal overgrowth, by a role for PHR in promoting directional axonal regrowth, or by a combination of limiting overgrowth and providing directional control. To test these possibilities, we determined the total regrown axonal length as the sum of rostrally and caudally regrown axons. The total axonal length did not significantly differ in *phr* mutants ($1023 \pm 462\,\mu m$, $n = 22$) and nonmutant siblings ($1082 \pm 349\,\mu m$, $n = 29$; Fig. 3m), suggesting that PHR has little effect on overall axonal regrowth. The most frequent misdirection of PHR-deficient axons occurred at the transection site. Therefore, we focused our further analyses on the directional regrowth across the transection site, i.e. the selection of the rostral or caudal path at the transection site. The excessive rostrally directed regrowth of PHR-deficient axons at the transection site could either be explained by PHR suppressing excessive multidirectional branching at the transection site or by PHR specifically suppressing rostrally misdirected regrowth. To distinguish between these possibilities, we counted the total number of caudally and rostrally directed branches at the transection site. PHR suppressed both caudally directed and rostrally misdirected branches (Fig. 3n). However, PHR suppressed almost five times more rostrally misdirected than caudally directed branches (Fig. 3n). Therefore, the percentage of rostrally directed branches was significantly increased in *phr* mutants (Fig. 3o). This suggests that PHR predominantly suppresses misdirected trajectories.

Next, we sought to determine how PHR promotes directional regrowth of CNS axons. To limit misdirected regrowth, PHR could either suppress the formation of misdirected processes or promote the removal of misdirected processes. To determine which of these two mechanisms is used by PHR to regulate misdirected regrowth across the transection site, we performed time-lapse imaging of regrowing Mauthner axons (Fig. 5) and determined the percentage of rostrally directed sprouts at 13, 18 and 23 hpt. In wild-type

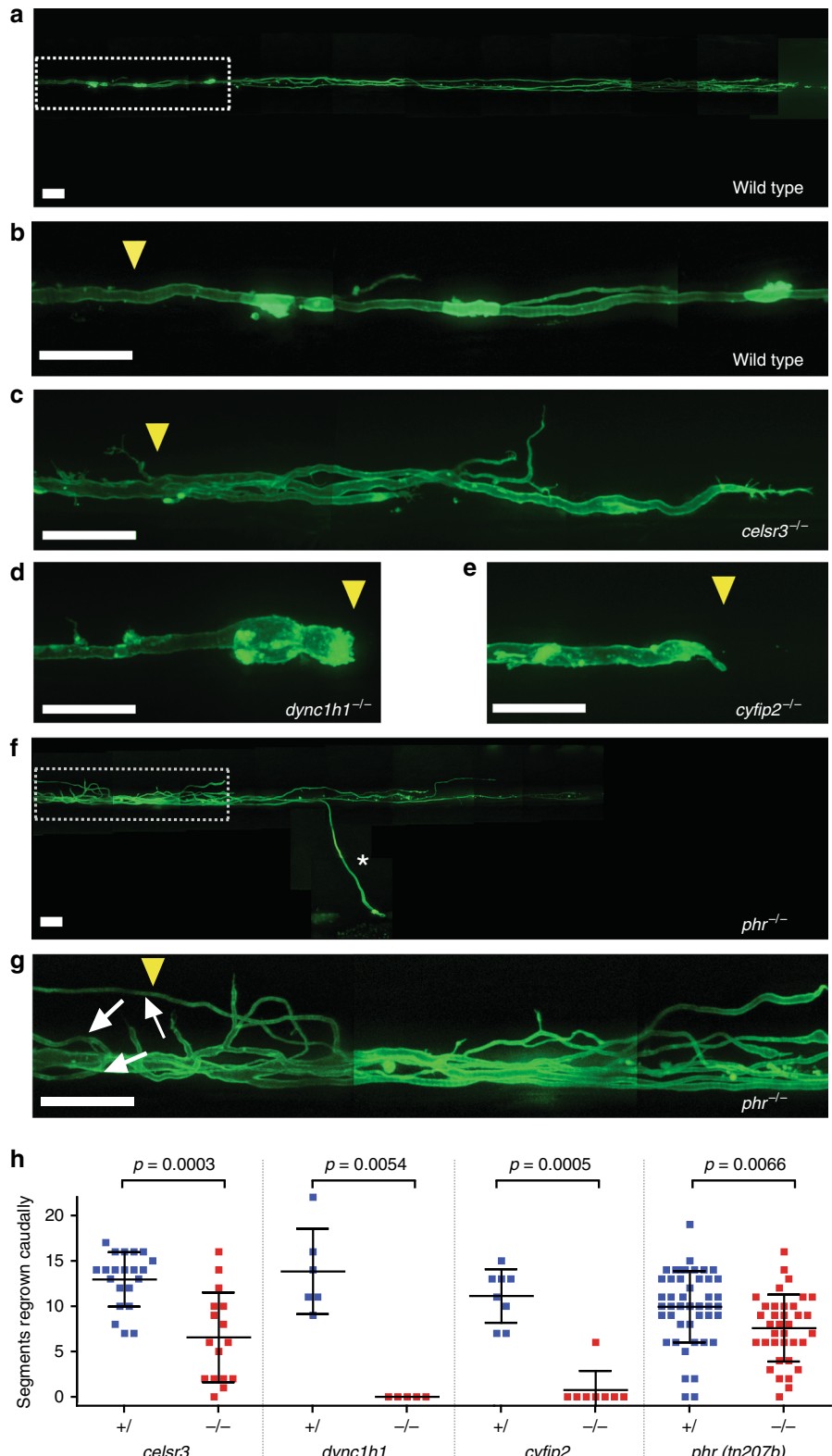

larvae, the percentage of rostrally directed sprouts decreased by 45% over time, indicating correction of misdirected processes. In *phr* mutants, however, the percentage of rostrally misdirected processes remained high (Fig. 5). Thus, PHR limits rostrally misdirected regrowth at least in part by correcting misdirected processes.

**PHR does not control MBP- and GFAP-positive glial morphology**. In adult zebrafish, GFAP-positive processes have been shown to guide regrowing axons across the transection site[50,51], while myelin expressed by oligodendrocytes is known to impair axonal regrowth in the mammalian CNS[1]. We therefore wondered whether PHR promotes directional axon regrowth by controlling the morphology of MBP- or GFAP-positive glial cells. Using time-lapse imaging, we analyzed the morphology of MBP- and GFAP-positive glial cells at and around the transection site during initial axonal regrowth (Fig. 6). We observed minimal

**Fig. 2** *Celsr3*, *dync1h1*, *cyfip2*, and *phr* are required for the extent of caudally directed Mauthner axonal regrowth. **a–h** Mauthner axons were labeled in double transgenic zebrafish larvae: *Tg(hspGFF62a)*, expressing Gal4 in the Mauthner combined with *Tg(UAS:gap43$_{1-20}$-citrine)*, driving expression of membrane-attached citrine. Extent of caudally directed Mauthner axonal regrowth at 48 hpt is shown. A stitched low magnification image of the entire length of a regrown wild-type axon is shown in (**a**). **b** is the zoomed-in image of the boxed area in (**a**), showing the regrown wild-type axon around and caudal to the transection site. At the same magnification as the regrown wild-type axon in (**b**), images of *celsr3*, *dync1h1*, and *cyfip2*-mutant axons at 48 hpt are shown (**c–e**), quantified in (**h**), demonstrating that *celsr3*, *dync1h1*, and *cyfip2* are required for the extent of Mauthner axonal regrowth. At the same magnification as the regrown wild-type axon in (**a**), an image of a regrown *phr* mutant axon around and caudal to the transection site is shown in (**f**) and a zoomed-in image of the boxed area in (**f**) is shown in (**g**), quantified in (**h**), demonstrating that PHR is also required for the full extent of caudally directed Mauthner axonal regrowth. An extraspinally regrown axon branch is marked by a white asterisk (**f**). In addition to the caudally regrown axon, several branches have regrown rostrally (white arrows), towards the neuronal cell body in the brain stem. Transection sites are marked by a yellow arrowhead. All scale bars are 30 μm. *P*-values in (**h**) were determined using two-tailed Student's *t*-test (*celsr3*, *phr*) or Mann–Whitney test (*dync1h1*, *cyfip2*). *N* = 21 nonmutant *celsr3* siblings, *n* = 16 *celsr3* mutants, *n* = 6 nonmutant *dync1h1* siblings, *n* = 5 *dync1h1* mutants, *n* = 8 nonmutant *cyfip2* siblings, *n* = 8 *cyfip2* mutants, *n* = 48 nonmutant *phr* siblings, *n* = 36 *phr* mutants were analyzed

damage to MBP- and GFAP-positive glial cells at the transections site in all larvae (Fig. 6). During the first ~9 h of axonal regrowth, when misdirected sprouts are corrected in wild-type larvae, we observed neither repair of glial cell damage nor glial scar formation in any of the analyzed larvae. Furthermore, the morphology of the examined glial cells in all wild type and *phr* mutants did not obviously differ. Therefore, PHR is unlikely to promote directional Mauthner axon regrowth by controlling glial cell morphology.

**PHR directs axon regrowth through *cyfip2*- and JNK-pathways.** Axons navigate through activity of the growth cone at the axon tip. Growth cone forward movement and stability of axonal processes are controlled by the cytoskeleton composed of a central microtubule core, F-actin-rich lamellipodia, and F-actin-rich protruding filopodia. We generated a transgenic zebrafish line to simultaneously monitor microtubule and F-actin dynamics live in regrowing Mauthner axons and determined whether PHR controls growth cone morphology. Using time-lapse imaging, we observed that PHR-deficient growth cones were significantly larger and had longer filopodia (Fig. 7, Supplementary Fig. 3, Supplementary Movies 3 and 4). In line with a possible cell-autonomous function of PHR in controlling growth cone size and filopodia length, antibody labeling indicated that PHR is expressed in Mauthner neurons (Supplementary Fig. 4). Since growth cone size and filopodia length are driven by F-actin polymerization, this observation raised the possibility that PHR limits F-actin polymerization. Therefore, we next tested whether *cyfip2*, which is known to promote F-actin polymerization[52–55] genetically interacts with *phr*. At 5 dpf, *cyfip2*-deficient axons almost completely failed to regrow after transection (Fig. 2). Since we neither observed any obvious defects in the morphology of MBP- and GFAP-positive glial cell nor any glial scar formation in *cyfip2* mutants before or after axonal transection (Supplementary Fig. 5), *cyfip2* is unlikely to act in glial cells to control axonal regrowth. To test the genetic interaction between *cyfip2* and *phr*, we performed the experiments one day earlier, at 4 dpf. When transected at 4 dpf, likely due to maternally deposited *cyfip2* transcript, *cyfip2*-mutant Mauthner axons were still able to regrow, but to a lesser extent than in nonmutant siblings (Fig. 8a–c). Compared to single mutants, co-ablation of *phr* and *cyfip2* in *phr*$^{-/-}$ *cyfip2*$^{-/-}$ double mutants further reduced the length of caudally directed Mauthner axonal regrowth, suggesting that *phr* and *cyfip2* act in parallel positive pathways to promote caudally directed regrowth (Fig. 8c). Remarkably, reduction of *cyfip2* normalized the misdirected regrowth observed in the absence of *phr* (Fig. 8a, b, d), suggesting that *phr* controls directional Mauthner axonal regrowth through a *cyfip2*-dependent pathway, possibly involving F-actin.

In addition to controlling directional Mauthner axonal regrowth through a *cyfip2*-dependent pathway, PHR is known to negatively regulate the MAPK pathway, by ubiquitinating Dlk, which in turn acts in part through c-Jun N-terminal kinase (JNK) in developmental axon outgrowth and in axonal regrowth after injury in drosophila and *C. elegans*[23,24,29]. We therefore tested whether pharmacologically inhibiting JNK would rescue misdirected regrowth in PHR-deficient Mauthner axons (Fig. 8e–h). Since the MAPK pathway involving JNK is essential for axonal regrowth[24,56,57] high doses of the JNK inhibitor SP600125 completely abolished Mauthner axonal regrowth in larval zebrafish (Fig. 8g). Exposing laser-transected *phr* mutant axons to a concentration of the JNK inhibitor that did not significantly affect the extent of caudally directed axonal regrowth, we observed a significant reduction in the percentage of misdirectedly regrown Mauther axons (Fig. 8e–h). Together, these findings suggest that PHR controls directional Mauthner axonal regrowth at least in part through *cyfip2*- and JNK-dependent pathways.

## Discussion

Here we combined laser-mediated axon transection with confocal time-lapse imaging and determined several genes promoting extent of CNS axon regrowth. Furthermore, we have identified an early phase of CNS axonal regrowth, i.e. shortly after regrowth begins. During this phase regrowing wild-type axons initiate multidirectional sprouting and then correct misdirected sprouts in a PHR-dependent manner. In fact, to our knowledge this finding is the first evidence that vertebrate PHR controls axon regrowth. In addition, *phr* is to our knowledge one of the first vertebrate genes known to promote directional CNS axonal regrowth across the transection site and along the pre-lesional trajectory. Based on the capacity to correct misdirected sprouts during axonal regeneration, PHR recapitulates to some extent its role in development. For example, PHR-mediated axon pruning has been described as a mechanism to correct erroneous axonal projections in developing drosophila neurons[58]. However, while PHR is known to control the late phase of axonal outgrowth during development, when outgrowth terminates at the distal axonal end and synapses form[12–14,58,59], PHR controls the early phase of axonal regrowth after injury. In summary, we have identified genes controlling Mauthner axon regrowth and characterized an early phase of axon regeneration during which incorrectly directed axon sprouts are corrected in a PHR-dependent manner.

We showed that PHR controls directional Mauthner axonal regrowth at least in part through *cyfip2*- and MAPK JNK-dependent pathways. While *cyfip2* has not been shown to interact with *phr* before, a downstream role for MAPK signaling has been studied intensively. To control developmental axon outgrowth

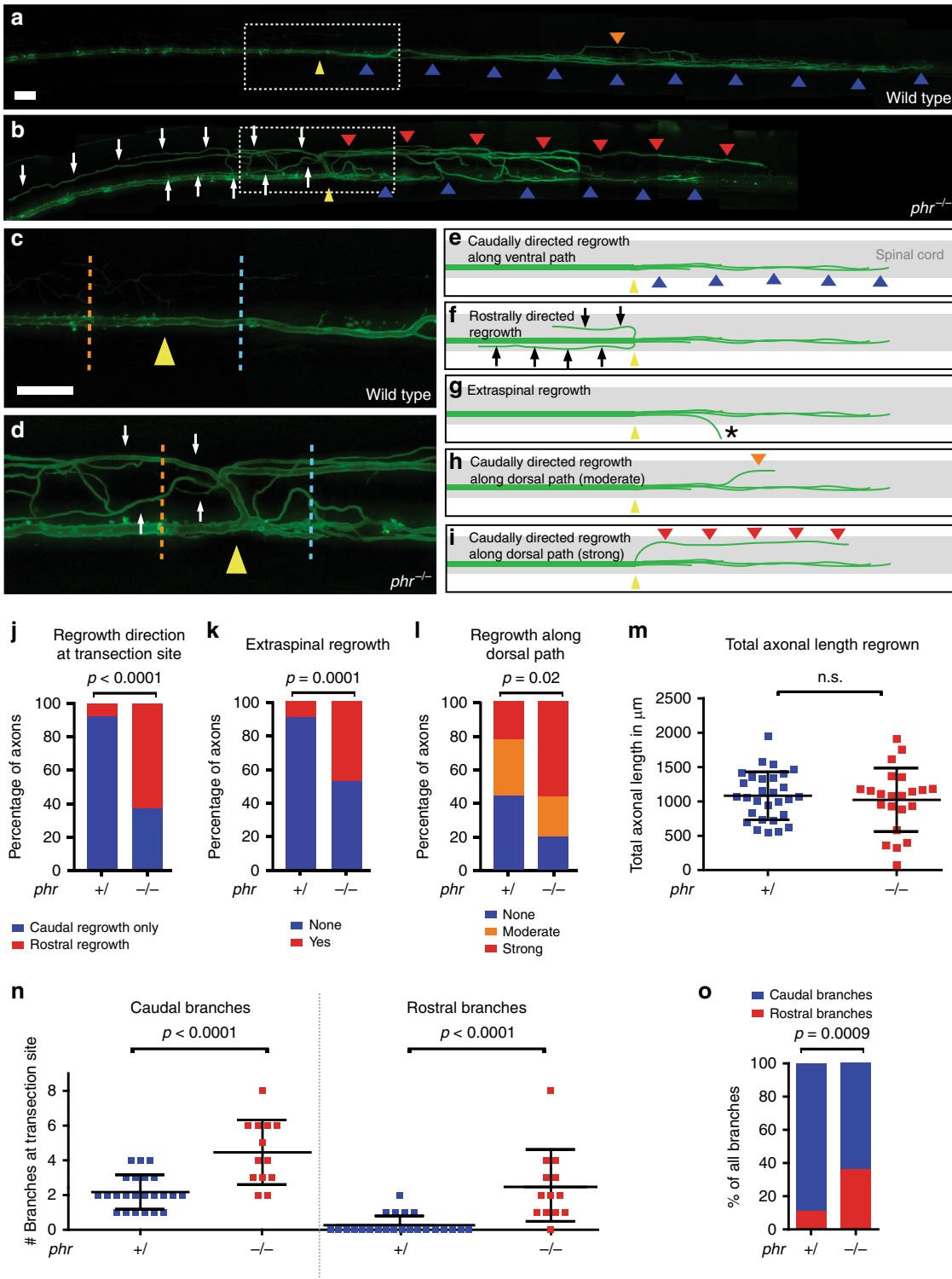

and regeneration in invertebrates, PHR has been shown to act through the MAP3K Dlk[19,20]. In developmental axonal outgrowth in vertebrates however, the role for Dlk is less clear. While Dlk has been reported not to be the main PHR effector in vertebrates in vivo[25,28], it has been shown to be regulated by PHR in cultured vertebrate neurons[18,48,60]. Downstream of MAP3K Dlk, PHR has been shown to act by regulating the MAPK p38 and JNK in axon regeneration in *C. elegans*[23] and to regulate JNK in developing drosophila and zebrafish neurons[29,58]. Hence, our finding that PHR controls directional Mauthner axonal regrowth

at least in part through a JNK-dependent pathway is in line with molecular downstream effectors of PHR in axonal development and invertebrate axonal regrowth.

In contrast, the genetic interaction between *cyfip2* and *phr* identified here has not previously been demonstrated. While *phr* and *cyfip2* act in parallel positive pathways to control the extent of axonal regrowth, ablation of *cyfip2* rescued directionality of axon regrowth in *phr* mutants. This suggests that depending on which aspect of regeneration is analyzed, there is a variable genetic relationship between *phr* and *cyfip2*. Cyfip2 is known to promote

**Fig. 3** PHR promotes directional regrowth of Mauthner axons. **a–d** Mauthner axon at 48 hpt; stitched, low magnification images of a wild type (**a**), and a *phr* mutant axon (**b**). Higher magnification of boxed areas (**a**, **b**) shown in **c** (wild type) and **d** (*phr* mutant). Yellow arrowheads mark the transection sites; white arrows mark rostrally directed projections in *phr* mutant. Caudally directed regrowth either labeled by blue arrowheads (along ventral spinal cord), orange arrowhead (moderate misdirected regrowth along dorsal spinal cord) or red arrowheads (strong misdirected regrowth along dorsal spinal cord). All Scale bars: 30 µm. **e–i** Schematic representation of Mauthner axonal regrowth: Caudally directed regrowth along ventral spinal cord (blue arrowheads; **e**), rostrally misdirected regrowth (black arrows; **f**), extraspinal regrowth (asterisk in **g**; see also Fig. 2f) or caudally misdirected regrowth along dorsal spinal cord (red/orange arrowheads; **h, i**). Quantification of regrowth direction at the transection site (**j**). Axons regrown exclusively in the caudal direction (blue bar); axons showing additional or exclusive rostrally directed regrowth (red bar) in *phr* mutants (*n* = 37) and nonmutant siblings (*n* = 27). Quantification of extraspinal ventral regrowth (**k**). Quantification of caudally directed regrowth along dorsal spinal cord (**l**) using a three-category rubric of "no", "moderate" or "strong" misdirected regrowth along dorsal spinal cord (see also **e, h, i**); *p*-values determined by Fisher exact tests (**j-l**). N = 21 *phr* mutants and *n* = 30 nonmutant siblings were analyzed (**k, l**). Quantification of total length of regrown axons (sum of rostral and caudal regrowth; in *n* = 22 *phr* mutants and *n* = 29 nonmutant siblings with *p*-values determined using two-tailed Student's *t*-test (**m**). Absolute number of rostral and caudal branches per axon in *phr* mutants (*n* = 13) and nonmutant siblings (*n* = 23), 40 µm rostral (orange dotted line; **c, d**), and 40 µm caudal (blue dotted line; **c, d**) to the transection site (**n**). Compared to nonnmutant siblings, caudally directed branches were increased twofold and rostrally directed branches were increased more than ninefold in *phr* mutants, (**n**); resulting in a higher percentage of rostrally directed branches in regrown *phr* mutant axons (**o**). *P*-values were determined using two-tailed Student's *t*-test (N, caudal), Mann–Whitney test (N, rostral) or Fisher exact test (**o**)

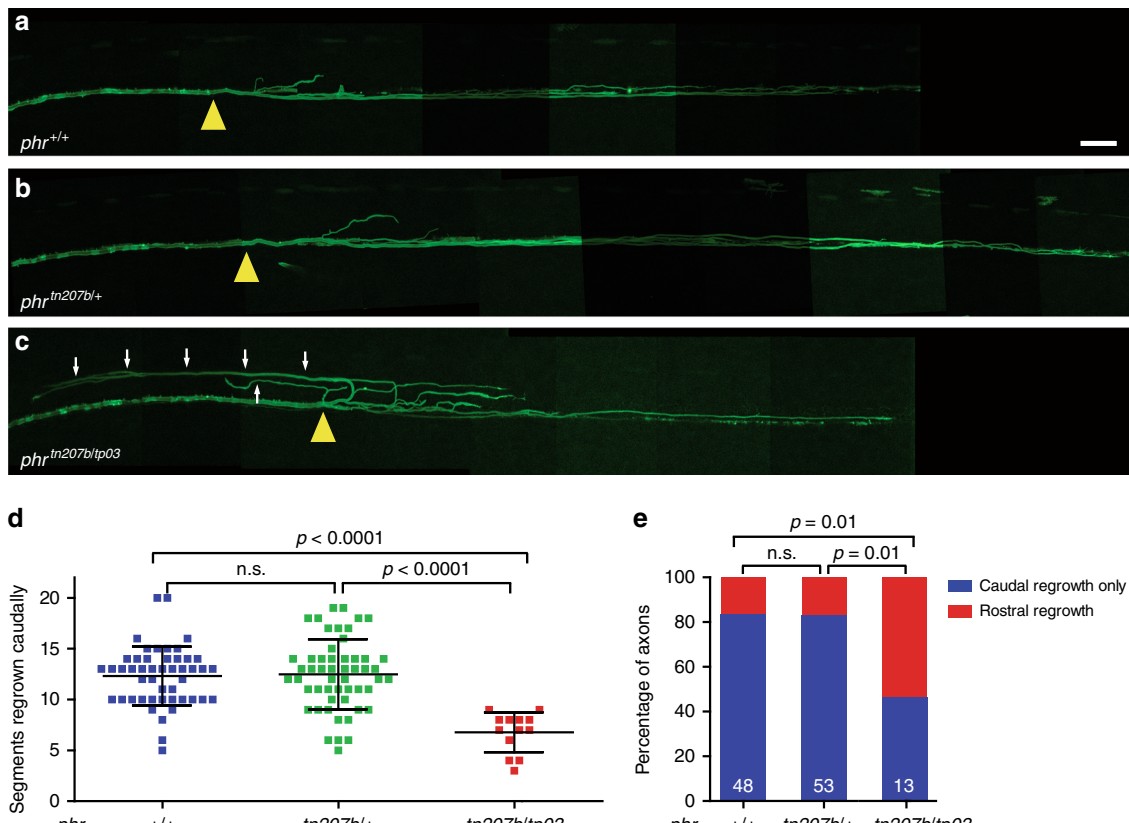

**Fig. 4** Mutation in *phr* is causative for the regeneration defect in *phr* mutants. **a–c** Mauthner axon at 48 hpt distal and proximal to the transection site in stitched, low magnification images of a wild-type *phr*$^{+/+}$ axon (**a**), a heterozygous *phr*$^{tn207b/+}$ axon (**b**) and a compound heterozygous *phr*$^{tn207b/tp03}$ axon (**c**). Transection sites marked by a yellow arrowhead; rostrally directed projections in the *phr*$^{tn207b/tp03}$ compound heterozygote are marked by white arrows (C). Scale bar for **a–c** : 30 µm. Quantification of the extent of caudally directed regrowth in number of segments regrown caudally (**d**) and quantification of regrowth direction at the transection site (**e**), showing normal extent and directionality of regrowth in *phr*$^{+/+}$ and *phr*$^{tn207b/+}$ but reduced extent and misdirected regrowth in compound heterozygous *phr*$^{tn207b/tp03}$. This shows that the regeneration deficits observed in *phr* mutants are indeed caused by a mutation in *phr*. N = 48 *phr*$^{+/+}$, *n* = 53 heterozygous *phr*$^{tn207b/+}$ and *n* = 13 compound heterozygous *phr*$^{tn207b/tp03}$ were analyzed in (**d, e**). *P*-values were determined using two-tailed Student's *t*-test (**d**), or Fisher exact test (**e**)

F-actin polymerization[52–55]. Since the effects of *phr* in directing Mauthner axonal regrowth were counterbalanced by *cyfip2*, it is possible that PHR directly or indirectly limits F-actin polymerization in the growth cones of regrowing CNS axons. During developmental axon outgrowth, PHR controls microtubule organization in growth cones in mouse[18], *C. elegans*[58], and zebrafish[29]. In addition to microtubule defects, abnormal F-actin-rich lamellipodia were described in *phr*-deficient murine motoneuron growth cones[18]. There is a cross-talk between actin and microtubules[61] as microtubules or microtubule-associated proteins can locally activate Rac1 and F-actin poymerization in nonneuronal and neuronal cells in vitro[62,63]. Hence, through the control of microtubules, PHR could indirectly control F-actin polymerization. Alternatively, it is possible that PHR controls F-

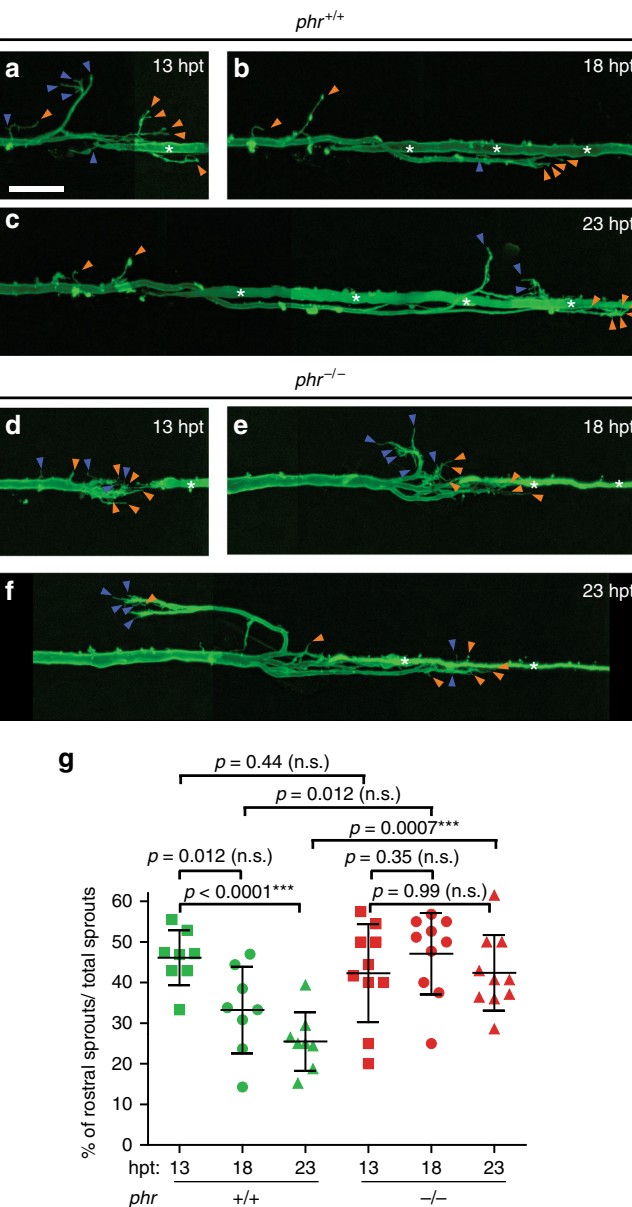

**Fig. 5** PHR is required to destabilize misdirected sprouts. Time-lapse imaging over 10 h of regrowing Mauthner axons in wild-type larvae and *phr* mutants, starting at 13 hpt. At 13, 18, and 23 hpt the regrown Mauthner axon of a wild-type larva (**a**–**c**) and a *phr* mutant (**d**–**f**) is depicted. At 13 hpt the regrowing Mauthner axons sprouted multidirectionally in both, wild type (**a**) and *phr* mutant (**d**). We quantified the number of all sprouts pointing either correctly caudally (more than 90° from the original proximal projection, a subset is labeled by orange arrowheads in **a**-**f**) or incorrectly rostrally (up to 90° from the original proximal projection, a subset is labeled by blue arrowheads in **a**-**f**). Quantification is shown in **g**. The data show that initially both *phr* mutant and wild-type axons displayed rostrally and caudally directed sprouts. While the percentage of rostrally directed sprouts remained high over time in *phr* mutants (**f**, **g**), the percentage decreased significantly in wild-type larvae at 23 hpt (**c**, **g**). Hence the percentage of rostrally directed sprouts was significantly higher in *phr* mutants at 23 hpt. This suggested that PHR directs regrowing axons, at least in part, by destabilizing misdirected sprouts. N = 10 *phr* mutants and n = 8 wild-type larvae were analyzed (one axon per fish). P-values were determined using two-tailed Student's t-test, p-values ≤ 0.007 indicated significant differences after Bonferroni correction. Scale bar in (**a**) is 30 μm

actin polymerization directly. To date, there is no evidence of a direct effect of PHR on F-actin[33]. However, PHR was found to bind and colocalize with F-actin in vitro[64] and to act in parallel with molecules that regulate the actin cytoskeleton, including Rho1 and Cdc42 in *C. elegans*[58]. We observed an increased size of *phr*-deficient growth cones in line with previous reports in *C. elegans*[58]. Furthermore, *phr*-deficient growth cones displayed longer filopodia. Filopodia protrusions are driven by F-actin polymerization and cannot form in the presence of inhibitors of actin polymerization. In addition, microtubules alone do not produce filopodia-like neurite extensions[65]. Therefore, the increased filopodia length, which we observed in PHR-deficient axons, could indicate that PHR enhances F-actin polymerization directly. Future work will be required to determine whether PHR directly controls F-actin polymerization in regenerating growth cones and to further the mechanistic understanding of how PHR controls directional Mauthner axonal regrowth through *cyfip2*- and JNK-dependent pathways.

Here we show that vertebrate PHR controls axonal regrowth after injury as has been shown in drosophila and *C. elegans*. However, although PHR plays an evolutionarily conserved role in controlling axonal regrowth, we observed substantial differences in how PHR controls axonal regrowth. First, we found that PHR promotes directional regrowth, which has not been reported in invertebrates. Second, whereas invertebrate PHR is known to limit axonal regrowth[22–24], we found that vertebrate PHR did not affect overall axonal regrowth (Fig. 3m). Instead, PHR directed axonal regrowth across the transection site and along the pre-lesional trajectory in zebrafish, at least in part, by controlling error correction of misdirected sprouts. Furthermore, while we observed a 37% delay in Wallerian degeneration in PHR-deficient peripheral nerves, PHR did not affect the onset or duration of Mauthner axon degeneration. Therefore, depending on which axon we studied, PHR had a variable effect on Wallerian degeneration which if present was rather mild compared to the strong effect of PHR on axon degeneration in mouse and drosophila[24,48]. Together this suggests that PHR variably controls axon de- and regeneration in a species- and axon-type dependent manner.

For two reasons, the distinct functions of PHR in controlling axonal regrowth in invertebrates and directional regrowth in vertebrates might not be entirely surprising. First, although the role of PHR in developmental axon guidance is also evolutionarily conserved, substantial differences in how PHR regulates axonal outgrowth have been reported. While developing axons in PHR-deficient invertebrates showed aberrant branching and overgrowth beyond the target[12,13,21,26], PHR-deficient axons were not only misguided but frequently stalled in mice[18] and zebrafish[27], demonstrating a vertebrate-specific, critical function of PHR at some choice points[27]. In line with this distinct role for PHR in development, PHR downstream targets[19,20,25,28,29] and functional consequences for the animal[17,25,33] have been shown to differ during development.

Second, axon growth and axon guidance are not completely independent processes and can be regulated by the same molecule. For example, in mouse neuropilin-2/class 3 semaphorin signaling limits developmental axon overgrowth in the hippocampus and prevents defasciculation of peripheral nerves[66]. In *C. elegans* PHR controls developmental axon branching and the extent of axon growth[13]. That the role of PHR in axon de- and regeneration differs depending on the type of neuron and/or on the species could possibly be explained by emerging evidence that PHR proteins can act as signaling hubs to regulate numerous downstream pathways[33]. Although our finding that PHR controls directional axonal regrowth in part through a JNK-dependent pathway is in line with a role of PHR in regulating the MAP3K

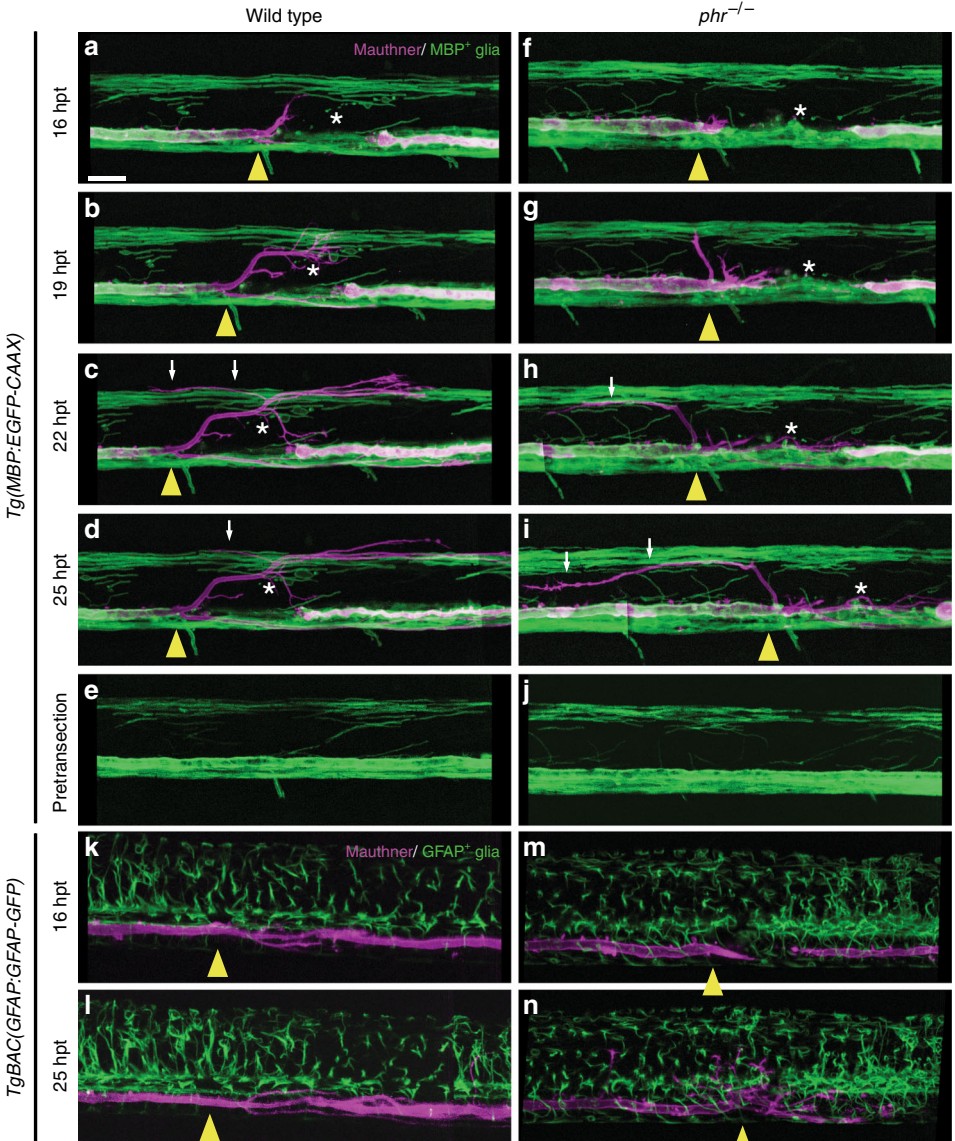

**Fig. 6** PHR does not control the morphology of MBP- and GFAP-positive glial cells. Time-lapse imaging over 9 h of regrowing Mauthner axons labeled by *Tg* (*hspGFF62a*) and *Tg(UAS:gap43$_{1-20}$-RFP)*. Myelinating oligodendrocytes are transgenically labeled in *Tg(MBP:EGFP-CAAX)* in (**a–j**) and GFAP-positive glial cells are labeled in *TgBAC(GFAP:GFAP-GFP)* in (**k–n**). In wild type (**a–e**) and *phr* mutant (**f–j**), laser-mediated axon transection caused minor damage (fragmentation and rounding) of myelinating oligodendrocytes (white stars) around the transection site (yellow arrowheads), compared to the pretransection images (**e, j**). Except minor damage, myelinating oligodendrocytes neither displayed obvious morphological changes nor interfered with the transection site or formed any obvious scar tissue in any of the larvae ($n = 4$ wild type and $n = 4$ *phr* mutants). We did not observe any differences in the morphology of MBP-positive glial cells between wild type and *phr* mutant larvae before or after transection. Note also the rostrally extending axonal processes in wild type (retracting over time; shorter in (**d**) than in (**c**), marked by white arrows), and in the *phr* mutant (longer in (**i**) than in (**h**), marked by white arrows). **k–n** Laser-mediated axon transection also caused minor damage to GFAP-positive glial cells in all larvae analyzed ($n = 4$ wild type, $n = 4$ *phr* mutants). Except minor damage, GFAP-positive glial cells neither displayed obvious morphological changes nor interfered with the transection site or formed any obvious scar tissue in any of the larvae ($n = 4$ wild type and $n = 4$ *phr* mutants). We did not observe any differences in the morphology of GFAP-positive glial cells between wild type and *phr* mutant axons before or after transection. Scale bar in (**a**) is 20 μm

Dlk[19,20], additional known downstream targets such as mTOR[27,28] or novel downstream pathways could contribute to the role of PHR in promoting directional CNS axonal regrowth.

Hence PHR has an evolutionarily conserved function in axonal regrowth with substantial differences in how PHR controls axonal regrowth in invertebrates and vertebrates. Future work is required to assess whether PHR is also able to promote directional regrowth of CNS axons in mammals and whether PHR and its downstream targets could contribute to a combinatorial treatment in spinal cord injury to direct axonal regrowth.

In addition to the role of PHR in promoting directional CNS axonal regrowth, we identified three genes that are required for the extent of vertebrate CNS axonal regrowth (*celsr3*, *dync1h1*, and *cyfip2*). Celsr3 has been shown to control developmental CNS axon outgrowth[67], but has to our knowledge not previously been implicated in axonal regrowth after injury. Dync1h1 is required for dynein function, which in turn mediates retrograde transport in neurons[68]. The essential binding partner of dynein, the dynactin complex, has previously been shown to be required for axonal regrowth in drosophila[24] and we have recently shown that

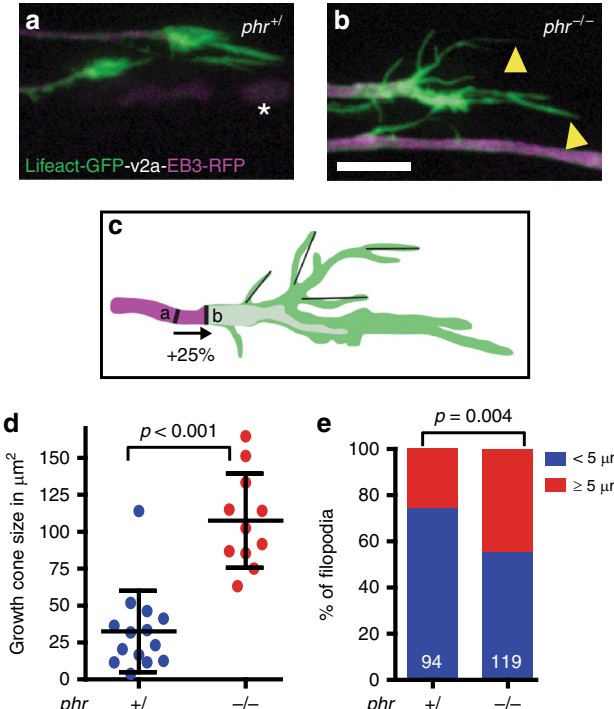

**Fig. 7** PHR controls growth cone size and filopodia length. **a–e** F-actin (green) and microtubules (magenta) were simultaneously labeled to visualize the growth cone cytoskeleton of regrowing Mauthner axons, in double transgenic Tg(hspGFF62a) Tg(UAS:lifeact-GFP-v2a-EB3-RFP) nonmutant siblings (**a**) or phr mutants (**b**) with long F-actin-rich filopodia labeled by yellow arrowheads (**b**); debris of the distal axon stump marked by white asterisks (**a**). Laser-mediated axon transection were performed in 5-day-old larvae and regenerating axons were imaged between 12 and 15 hpt. Scale bar in (**b**) is 10 μm. Schematic drawing of the phr mutant growth cone, showing how the growth cone was defined and how filopodia length was measured (**c**). The beginning of the growth cone was defined as an increase in the axon diameter of 25% or more compared to the proximal axon shaft, which usually correlated with an obvious increase in F-actin labeling. Filopodia were measured by a straight line from filopodia base to the end of the visible F-actin signal. Growth cone size as the area above a defined intensity threshold is displayed in (**d**). Growth cones were significantly larger in regrowing phr mutant axons compared to nonmutant siblings. P-value was determined using two-tailed Student's t-test. Filopodia length was determined and the percentage of filopodia below or ≥ 5 μm shown as bar graphs (**e**). Significantly more filopodia ≥5 μm were seen in regrowing phr mutant axons compared to nonmutant siblings. P-value was determined using the Fisher exact test. N = 14 growth cones in 5 non-mutant siblings and n = 11 growth cones in 5 phr mutants larvae were measured; n = 94 filopodia in 5 nonmutant siblings and n = 119 filopodia in 5 phr mutant larvae were measured. More growth cones and individual channels are shown in Supplementary Fig. 3

dynein heavy chain is required for the extent of peripheral axon regrowth in larval zebrafish[69]. In fact, a role for dynein in retrograde injury signaling and in axonal regeneration is now widely accepted[70]. Cyfip2 as a component of the WAVE-regulatory complex regulates F-actin polymerization and has been shown to control developmental axon guidance[71,72]. However, a role for Cyfip2 in axon regeneration has, to our knowledge, not been previously reported. Thus, of the 15 genes tested, we have identified four that control vertebrate CNS axonal regrowth, indicating that laser-mediated Mauthner axon transection is a powerful tool to determine the requirements for CNS axonal regrowth. It remains to be determined whether a failure to

express Celsr3, Dync1h1 and/ or Cyfip2 contributes to the failure of mammalian CNS axons to regrow after injury.

## Methods

**Zebrafish care and strains.** Protocols and procedures involving zebrafish (Danio rerio) are in compliance with the University of Pennsylvania Institutional Animal Care and Use Committee regulations. Embryos were generated by natural mating as described[73]. Embryos were raised at temperatures between 25 and 28 °C. This study was performed using larval zebrafish. The sex in larval zebrafish is not yet determined. To label the Mauthner neuron, we used an enhancer trap line expressing Gal4 in Mauthner neurons Tg(hspGFF62a)[38] in combination with Tg (UAS:gap43₁₋₂₀-citrine), also called p210Tg, expressing the first 20 amino acids of GAP43 fused to citrine[74] or Tg(UAS:gap43₁₋₂₀-RFP), which we generated by fusing the first 20 amino acids of GAP43 to RFP. To label F-actin and microtubules simultaneously, we generated Tg(UAS:lifeact-GFP-v2A-EB3-RFP) fish using Tol2 transgenesis[75], expressing GFP-tagged F-actin-binding lifeact[76] and RFP-tagged human EB3, both separated by self-cleaving viral 2A peptide. Four genes controlled Mauthner axonal regrowth, the following alleles were tested: cyfip2$^{p400}$ mutants were generated and identified as previously described[77], celsr3$^{fh339}$ mutants were kindly provided by C. Moens[78], dync1h1$^{hi3684Tg}$[79]. For mycbp2, here called phr we used the tn207b allele, also referred to as esrom$^{tn207b}$[15] and the tp03 allele, also known as tp203[15]. Ten genes and the combination of nf1a and nf1b did not affect Mauthner axonal regrowth. The following alleles were tested: cfl1l $^{sa5863}$[80], dicer1$^{hu896}$[81], dgcr8$^{fh344}$ kindly provided by C. Moens[78], slow learner$^{p174}$[82], lrp4$^{p18483}$, nf1a$^{Δ5}$ and nf1b$^{+1032}$, pappaa$^{p170}$[82], rb1$^{te226a}$[84], robo2$^{ti272z}$[85], sox10$^{m241}$[86], and tnc$^{sa1576}$[80]. To label GFAP-positive glia cells, we used the transgenic line tgBAC(GFAP:GFAP-GFP)$^{zf167tg}$[87] and to label MBP-positive glia cells, we used the transgenic line tg(MBP:EGFP-CAAX)$^{ue2tg}$[88]. To label motor axons in peripheral nerve, we used the transgenic line Tg(mnx1:GFP)$^{ml2}$[89]. Larvae used for experiments were selected based on transgene expression and when present on genotype-dependent morphology, i.e. morphological phenotype of mutant larvae. If applicable (e.g. pharmacological treatment), larvae were randomly distributed between experimental and control groups.

**Mauthner axon and peripheral nerve transection.** Mauthner axons were transected at the ninth hemisegment and peripheral nerves were transected as previously described[90,91]. In brief, live larvae were mounted in 1.2% agarose in E3 and anesthetized in 0.01% tricaine. Mauthner axons or peripheral nerves were then visualized on a spinning disc confocal microscope. We used a pulsed nitrogen or a solid state laser on a spinning disc confocal microscope to transect the Mauthner axon or peripheral nerves in a defined region of interest containing almost exclusively the axon/nerve. Successful transection was assessed 15–60 min post transection for the Mauthner axon or ~10 hpt for peripheral nerves. For all conditions tested, we first performed a pilot experiment with a minimum of five larvae per genotype/group. In some instances this already resulted in significant regeneration defects. If a nonsignificant trend toward a regeneration defect was noticed, the sample size was increased until it was clear whether there was a real or a random difference. No statistical method was used to predetermine sample size, because we could not predict whether there would be a regeneration defect or the type or severity of the defect. Most mutant larvae tested had a morphological phenotype and were preselected from clutches based on phenotype. Since larvae had to be mounted under a stereo microscope, they were visually inspected before confocal microscopy, a procedure which cannot be blinded. Therefore blinding was not possible during laser-mediated transections and posttransection microscopy. However, the same transection parameters were applied to all larvae, independent of genotype/phenotype. Image analyses were performed on images with number codes and no associated genotype information (blinded). Genotype information was revealed after quantification. For details on data analysis see section on "Image acquisition, processing and data analysis".

**JNK inhibitor treatment.** Following laser-mediated Mauthner axon transection, larvae were incubated in E3 medium with a final concentration of 1% DMSO with or without addition of the JNK inhibitor SP600125 (Cell Signaling Technology) at different concentration as indicated in the result section. Larvae were kept at 28 °C until regeneration was assessed on a spinning disc confocal microscope. Ten millimolar stocks of the JNK inhibitor SP600125 were prepared in DMSO and kept at −20 °C.

**Time-lapse imaging.** Live larvae were mounted in 0.8% agarose in E3, anesthetized in 0.006% tricaine, and kept at 28 °C. Live larvae were imaged using a spinning disc confocal microscope. For Fig. 5, after transection, larvae were kept at 28 °C for 13 h before time-lapse imaging was started and then imaged every 5 h (three times), and for Fig. 6, larvae were kept at 25 °C for 16 h before time-lapse imaging was started and then imaged every 3 h (four times).

**Antibody labeling.** Larvae were fixed in 4% PFA in PBS with 0.25% Triton X100 (PBT) over night at 4 °C. After washing three times in PBT for at least 5 min, we performed antigen retrieval: 5 min incubation in 150 mM Tris HCl pH 9.0 at room temperature, followed by 15 min incubation at 70 °C. After washing twice in PBT for

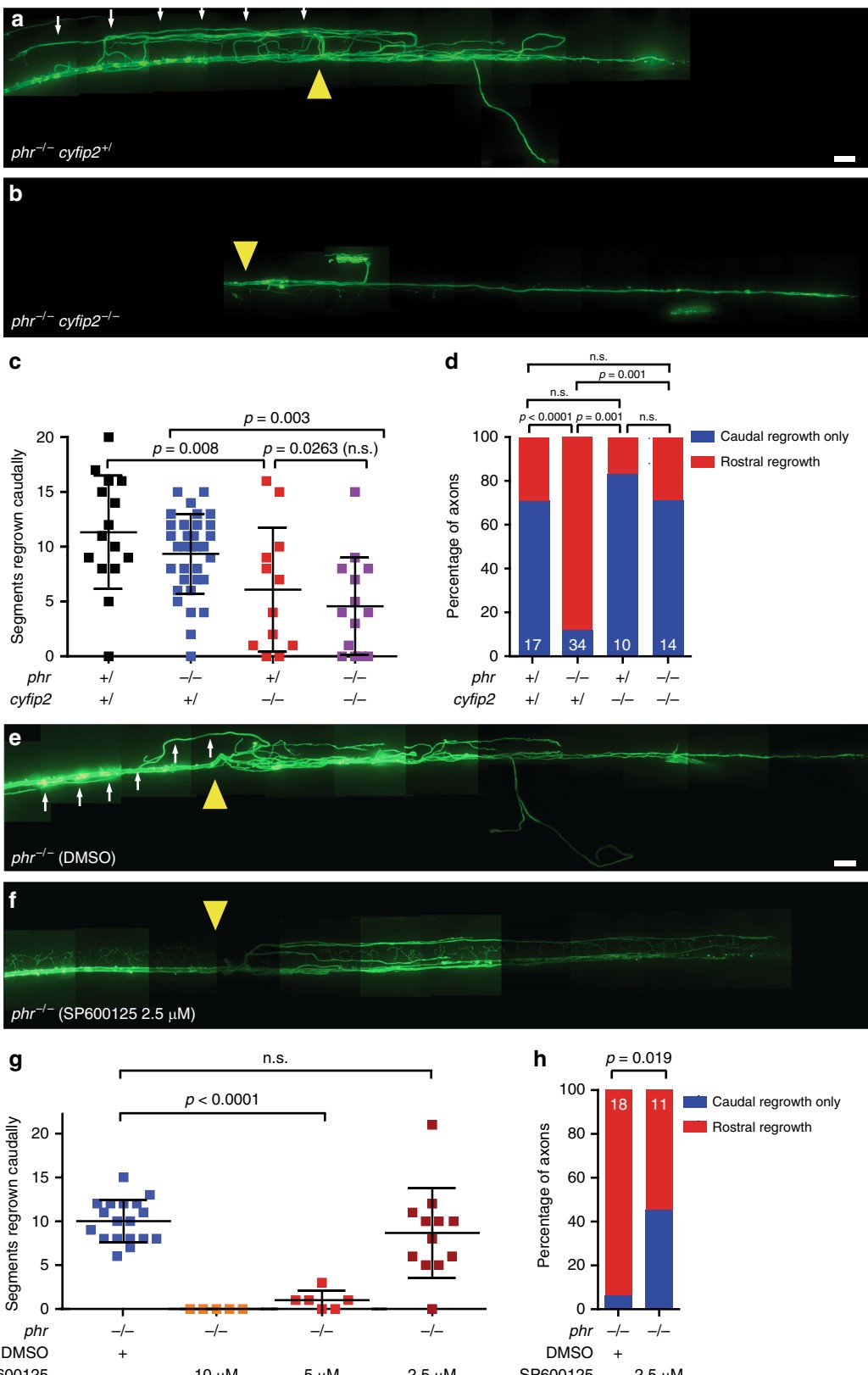

at least 5 min, we permeabilized the larvae by incubation in 0.05% Trypsin-EDTA on ice for 50 min. After washing three times in PBT for at least 10 min, the samples were blocked for 1 h in 2% normal goat serum, 1% BSA, and 1% DMSO in PBT.

To immunolabel PHR, we used a rabbit polyclonal antibody directed against human PHR (aa 4519–4534), called PP1[92]. Anti-PHR antibody (PP1) was kindly provided by Vijaya Ramesh and Roberta Beauchamp. Using the anti-PHR antibody, we observed a mainly nuclear but also cytoplasmic signal in the soma of

Mauthner neurons as well as a punctate staining along the axon. This distribution is in line with the localization previously reported for PHR[18,27,92]. Using protein BLAST (NIH), we found that parts of the antigenic peptide sequence share ≥80% similarity with three different regions of zebrafish PHR. PP1 was diluted 1:20 in 1% BSA, 1% DMSO in PBT and incubated for at least 36 h at 4 °C. After washing three times in PBT for at least 10 min, incubation with an Alexa-594-anti-rabbit secondary antibody was performed at a 1:400 dilution.

**Fig. 8** *Cyfip2*-deficiency and JNK inhibition rescue misdirected regrowth in *phr* mutant Mauthner axons. **a–d** Laser-mediated axon transections were performed in 4-day-old larvae with different *cyfip2* and *phr* genotypes. Regrowth was assessed 48 hpt. A stitched image of a regrown *phr* mutant axon is shown in (**a**). A yellow arrowhead marks the previous transection site. From the transection site several projections have regrown rostrally (white arrows). In contrast, in a $phr^{-/-}$ $cyfip2^{-/-}$ double mutant, there is only caudally directed regrowth (**b**). Scale bar: 30 μm. Quantification of extent of caudally directed regrowth (**c**). Quantification of percentage of axons with rostrally misdirected regrowth (**d**). P-value was determined using two-tailed Student's *t*-test (**c**) or Fisher exact test (**d**). N of axons (1 axon per larva) for (**c**) and (**d**) as indicated in white numbers in the bar diagram in (**d**), except for $phr^{+/}$ $cyfip2^{-/-}$: 12 axons were analyzed in (**c**) and 10 in (**d**), because two did not regrow at all in either direction. Only p-values ≤ 0.017 indicated significant differences after Bonferroni correction in (**c**). **e–h** Laser-mediated axon transections were performed in 5-day-old $phr^{-/-}$ larvae. After transection, larvae were treated with DMSO (control) or with JNK inhibitor SP600125. Regrowth was assessed at 48 hpt. Stitched images of regrown *phr* mutant axons are shown (**e**, **f**), DMSO control (**e**), JNK inhibitor SP600125 2.5 μM treated (**f**). Yellow arrowheads mark previous transection sites. From the transection site, there is a rostral trajectory (white arrows) in the DMSO-treated larva. Scale bar: 30 μm. Quantification of extent of caudally directed regrowth (**g**). SP600125 dose-dependently reduced Mauthner axonal regrowth. At 2.5 μM extent of Mauthner axonal regrowth was not significantly reduced. P-values were determined using two-tailed Student's *t*-tests (**g**). Quantification of percentage of axons with rostrally misdirected regrowth in DMSO- or 2.5 μM SP600125-treated larvae (**h**). Number of axons (one axon per larva) in (**g**): DMSO: $n = 18$; SP600125 10 μM: $n = 5$; 5 μM: $n = 6$; 2.5 μM: $n = 12$. Numbers in (**h**): DMSO: $n = 18$ and SP600125 2.5 μM: $n = 11$ (one less than in (**g**) because one did not regrow at all in either direction)

**Image acquisition, processing and data analysis**. Microscopic images were acquired using Slidebook (3i) or VisiView (Visitron) software. Maximum intensity projection images of z-stacks were created using Slidebook (3i) or Image J software package (NIH) software. Confocal images were further processed using Image J or FIJI. Image manipulations included adjustment of brightness, contrast, gamma-value, and background subtraction. Manipulations were always applied to the entire image and to all images in one experiment, ensuring that the content of the image was not altered. Images were exported and further processed in Photoshop CS4. In order to generate an image of the entire regrown axon, we stitched individual images together in Photoshop CS4. Final versions of the figures for the manuscript were prepared using Illustrator CS4 and Photoshop CS4 (Adobe). Extraspinal regrowth was determined at the time of microscopy, by determining the green fluorescent signal compared to brightfield images in which the border of the spinal cord can be discerned. For quantification of caudally directed regrowth along the dorsal spinal cord, we used a three-category rubric of no, moderate or strong misdirected regrowth along the dorsal spinal cord. Caudally directed dorsal regrowth was defined as growth in the dorsal half of the spinal cord; "no" indicates absent growth in the dorsal half of the spinal cord, "moderate" refers to regrowth in the dorsal half, but less than in the "strong" category. "Strong" corresponds to axonal regrowth in the dorsal half covering more than half of the caudally regrown axon length. One mutant axon was excluded from this quantification, because it showed almost no regrowth beyond the transection site. For scoring the extent of regrowth, we either counted the length of regrowth in number of hemisegments. Alternatively, we imaged the entire regrown axons, stitched the images and measured the length of the regrown axon. To measure the growth cone size, beginning of the growth cone was defined as an increase of axonal diameter by at least 25%, which correlated with an increase in F-actin labeling. Filopodium length was measured by drawing a straight line from the beginning at the lamellipodium of the growth cone to the end of the visible filopodia tip. To determine the onset of Wallerian degeneration, either time-lapse imaging of one section of the Mauthner axon at the anal level or of one entire peripheral nerve was performed and a precise time of axon fragmentation could be given. In order to compare Mauthner axon degeneration in *phr* mutants to nonmutant siblings, we visually inspected the entire length of the Mauthner axon at 17 or 48 hpt on a spinning disc confocal and scored fragmentation as complete, partial or none. Extent of peripheral nerve regrowth was quantified using the following three categories: no/weak, moderate, strong, as previously described[90,93]. No custom computer codes or algorithms were used in this study.

**Statistics and reproducibility**. *P*-values were calculated by the Fisher exact test for categorical data using a Graph Pad web tool (GraphPad), by the Student's *t*-test for normally distributed, numerical data, or by the Mann–Whitney test for not normally distributed numerical data using Graph Pad Prism 5 software (GraphPad). Graphs were generated using Prism 5 (GraphPad). If multiple testing was performed (Figs. 5 and 8), *p* values indicating significance were adjusted using Bonferroni correction. All error bars show the standard deviation. All attempts at replication were successful. Source data to generate graphs and 95% confidence intervals are provided in Supplementary Data 1.

**Reporting summary**. Further information on research design is available in the Nature Research Reporting Summary linked to this article.

## Data availability
We declare that all the data supporting the findings of this study are available within the paper and the supplementary information files. Source data used to generate the plots are included in Supplementary Data 1.

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

## Acknowledgements

We would like to thank members of the Granato lab for helpful discussions and comments on the project and the manuscript. J.B. was funded by a research fellowship from the German Research Foundation (DFG; grant number BR4680/1-1). M.G. received funding from the National Eye Institute (grant number R01 EY024861) and from the National Institute of Neurological Disorders and Stroke (grant number R56 NS097914). Anti-PHR antibody was kindly provided by Vijaya Ramesh and Roberta Beauchamp. Furthermore, we would like to thank Sabine Goergens, Francesca Peri, Werner Boll, and Ambra Villani for their support during the revision of the manuscript and Adriano Aguzzi and Elisabeth Rushing for constructive comments on the manuscript.

## Author contributions

J.B. and M.G. conceptualized this work and acquired funding. M.G. provided resources and supervised the work. J.B. performed the experiments, analyzed the data and wrote and revised the manuscript. A.M. and K.M. provided zebrafish lines, helped in the conceptualization of this study and revised the manuscript. All authors reviewed the manuscript.

## Additional information

**Competing interests:** The authors declare no competing interests.

