## [Peer Review File · Communications Biology]

Editorial Note: This manuscript has been previously reviewed at another journal. This document only contains reviewer comments and rebuttal letters for versions considered at *Communications Biology* .

REVIEWERS' COMMENTS:

Reviewer #1 (Remarks to the Author):

As noted in our previous review, the authors presented an interesting study. The revisions to the manuscript have greatly improved the manuscript! I would like to commend the effort of the authors to address each our suggested revisions. The revisions are elegant and make for a more complete story.

I would like to compliment the authors on the elegant experiment and quantification described in Figure 5, especially page 10, line 201 and Figure 6.

Reviewer #2 (Remarks to the Author):

This manuscript by Bremer and colleagues provides the first evidence that Phr/Esrom regulates axon regeneration in vertebrates. Moreover, the authors provide evidence that Phr affects filopodia and actin polymerization to impact the orientation and extent of axon regeneration.

The authors now include important new data on transheterozygotes with two Phr alleles (tp207b/tp203). A robust, quantitative data set indicates that transheterozygous animals have similar defects in regeneration as homozygous tp207b homozygous mutants. Inclusion of this new data for a second Phr allele substantially bolsters my confidence in the overall findings of this manuscript and greatly strengthens the authors' conclusion that Phr regulates axon regeneration in zebrafish.

The authors have also entirely addressed my other concerns. This revised manuscript now has my full support. However, there are a couple of minor points that should be addressed, but I am happy to have the editor and authors handle this without further review.

- 1) One minor suggestion. It would be helpful to add the Phr allele information to Figure 2 just to make sure the reader can easily tell which allele is used for the majority of the experiments.
- 2) In Figure 8C, the comparison between *phr*^{-/-} *cyfip2*^{-/-} double mutants and *cyfip2*^{-/-} single mutants is a $p=0.0263$ but this is designated as not significant. Is this correct?
- 3) The authors state, "While *phr* and *cyfip2* act in parallel positive pathways to control the extent of axonal regrowth, ablation of *cyfip2* rescued the regrowth defect of *phr* mutants." should the second half of this sentence refer to defects in "direction" or "directionality" of regrowth in *phr* mutants?

Reviewer #3 (Remarks to the Author):

This revised manuscript is now significantly improved. I am satisfied with the additions and revisions the authors have made and I agree it is now ready for publication in *Communications Biology*.

Response to Reviewers

Reviewer #2

We agree and we have added the allele name to Figure 2.

As we state in the figure legend, p-values were determined using student's t-test. Since the graph includes multiples comparisons, we have performed a Bonferroni correction. After the Bonferroni correction, only p-values ≤ 0.017 indicate significant differences.

We agree and we have modified the sentence in the Discussion section to "While *phr* and *cyfip2* act in parallel positive pathways to control the extent of axonal regrowth, ablation of *cyfip2* rescued directionally of axon regrowth in *phr* mutants".

We would like to thank all three reviewers for their positive assessment of the revised version of our manuscript.